# Reduction-Hypersensitive Podophyllotoxin Prodrug Self-Assembled Nanoparticles for Cancer Treatment

**DOI:** 10.3390/pharmaceutics15030784

**Published:** 2023-02-27

**Authors:** Xinhui Wang, Yuequan Wang, Jiaxin Yu, Qian Qiu, Rui Liao, Shenwu Zhang, Cong Luo

**Affiliations:** Department of Pharmaceutics, Wuya College of Innovation, Shenyang Pharmaceutical University, Shenyang 110016, China

**Keywords:** podophyllotoxin, chemotherapy, prodrug, tumor-specific response, nanodrug delivery system

## Abstract

Podophyllotoxin (PPT) has shown strong antitumor effects on various types of cancers. However, the non-specific toxicity and poor solubility severely limits its clinical transformation. In order to overcome the adverse properties of PPT and explore its clinical potential, three novel PTT−fluorene methanol prodrugs linked by different lengths of disulfide bonds were designed and synthesized. Interestingly, the lengths of the disulfide bond affected the drug release, cytotoxicity, pharmacokinetic characteristics, in vivo biodistribution and antitumor efficacy of prodrug NPs. To be more specific, all three PPT prodrugs could self-assemble into uniform nanoparticles (NPs) with high drug loading (>40%) via the one-step nano precipitation method, which not only avoids the use of surfactants and cosurfactants, but also reduces the systemic toxicity of PPT and increases the tolerated dose. Among the three prodrug NPs, FAP NPs containing α-disulfide bond showed the most sensitive tumor-specific response and fastest drug release rate, thus demonstrating the strongest in vitro cytotoxicity. In addition, three prodrug NPs showed prolonged blood circulation and higher tumor accumulation. Finally, FAP NPs demonstrated the strongest in vivo antitumor activity. Our work will advance the pace of podophyllotoxin towards clinical cancer treatment.

## 1. Introduction

Cancer is a serious threat to human health [1]. At present, there are many choices for cancer treatment, such as surgical resection, chemotherapy, phototherapy, immunotherapy and gene therapy [2,3,4,5,6,7]. In addition to surgical resection, chemotherapy is the most commonly used remedy because of its advantages of antitumor broad-spectrum and systemic therapeutic effects [8,9]. However, traditional chemotherapeutic drugs also have some flaws, such as poor specificity, narrow treatment windows and insufficient tumor accumulation, which seriously affect the prognosis of treatment [10,11]. 

Podophyllotoxin (PPT) has been widely reported in many studies because of its potent antitumor activity [12,13,14,15,16,17]. It stands out among the many active ingredients of Podophyllum hexandrum Royle due to its strongest cytotoxicity [18]. However, there is still no precedent for PPT to be used in cancer clinical treatment due to its poor water solubility and severe non-specific cytotoxicity [19,20,21]. Therefore, it makes sense to develop novel delivery strategies that can stride over the PPT’s hurdle.

Nano-drug delivery systems (Nano-DDSs) have attracted much attention in recent years [22,23,24,25,26,27]. Nano-DDSs endow drugs with many preponderances: (i) improving the drug solubility [28,29,30,31]; (ii) prolonging the blood circulation time [32,33,34,35]; (iii) increasing the tumor accumulation [6,36,37,38]. However, only using the nano delivery strategy to deliver highly toxic chemotherapeutic drugs may cause more serious damage to normal organs and tissues due to the prolonged retention time in vivo [39,40]. A prodrug strategy may be a better choice for the delivery of highly toxic chemotherapeutic drugs, which refers to a compound that has low or no activity after modification but can release the active component to take effect at the target sites [41,42,43,44]. Thus, prodrug-based nanodrugs have emerged as a promising drug delivery platform due to the combination of features and advantages of prodrug strategies and nanodrug delivery technologies. Furthermore, carrier-free prodrug self-assembled nanodrugs overcome the shortcomings of traditional carrier-based nanomaterials to a certain extent and show unique advantages, such as simple preparation, high drug loading and negligible carrier-related toxicity.

Numerous previous studies have demonstrated that the antitumor effect of prodrug self-assembled nanodrug delivery systems depends on high tumor accumulation and rapid drug activation. In our previous study, it was found that the stronger the ability of the prodrugs to assemble, the longer the circulation and then the higher the tumor accumulation. Therefore, how to improve the self-assembly capacity of small molecule prodrugs has been the direction of our efforts. Many studies have shown that the introduction of 9-fluorenylmethoxycarbonyl (Fmoc) into nanocarriers can improve the drug loading capacity, which is attributed to the fact that Fmoc can be used as a functional building block to interact with drug molecules. Hence, we wonder if the introduction of Fmoc into small-molecule prodrugs can facilitate the nanoassembly of a prodrug. In addition to excellent assembly capacity, rapid drug activation also determines the ultimate antitumor effect. Disulfide bonds have been shown to respond specifically to high GSH at tumor sites, thus accelerating drug release. More importantly, the position of disulfide bonds in the linker also affects the redox responsiveness of the prodrug. 

To verify the above hypothesis, we designed three novel PPT-Fmoc prodrugs (α-Fmoc-SS-PPT, β-Fmoc-SS-PPT, γ-Fmoc-SS-PPT) linked by various lengths of disulfide bonds, abbreviated as FAP, FBP and FGP, respectively. Surprisingly, the three prodrugs could self-assemble into uniform NPs (abbreviated as FAP NPs, FBP NPs and FGP NPs, respectively) by introducing the Fmoc group, demonstrating the critical role of the Fmoc group in prodrug self-assembly (Figure 1). In addition, all three prodrug NPs showed higher cell uptake efficiency, better pharmacokinetic behavior and higher tolerant doses than PPT solution. However, the three prodrug NPs showed vastly different drug release capabilities affecting cytotoxicity and eventual in vivo drug activation. Finally, FAP NPs with an α-disulfide bond showed the most outstanding antitumor effect, which was attributed to higher reductive responsiveness and faster tumor-specific drug release than FBP NPs and FGP NPs. In conclusion, our study developed novel PPT prodrug self-assembled NPs for efficient cancer treatment, which provides a new possibility for the antitumor clinical practice of PPT.

## 2. Materials and Methods

### 2.1. Material

PPT, RPMI 1640 cell culture medium, DMEM-F12 cell culture medium, 3-(4,5-dimthyl-2-thiazolyl)-2,5-dipphenyl-2H-terazolium bromide (MTT) and Hoechst 33342 were purchased from Dalian Meilun Biotech Co., Ltd., Dalian, China. 9-Fluorene methanol was purchased from Shanghai Aladdin Biochemical Technology Co., Ltd., Shanghai, China. Dithiodiglycolic acid, 3,3′-dithiobispropionic acid and 4,4′-dithiodibutyric acid were obtained from TCI (Shanghai) Chemical Industry Development Co., Ltd., Shanghai, China. DSPE-PEG_2k_ was purchased from AVT (Shanghai) Pharmaceutical Technology Co., Ltd., Shanghai, China. Penicillin–streptomycin and fetal bovine serum were obtained from GIBCO, Invitrogen Corp., Waltham, MA, USA. Other chemicals and solvents were of analytical or high-performance liquid chromatography (HPLC) grade.

### 2.2. Synthesis of Prodrugs

FAP, FBP and FGP were obtained through two-step esterifications. In order to promote the esterification activity of dithioglycolic acid, 3,3′-dithioglycolic acid and 4,4′-dithioglycolic acid, they were stirred at room temperature for 2 h in the system of acetic anhydride, and then, dithiodiacetic anhydride, 3,3′-dithiodipropionic anhydride and 4,4′-dithiodibutyric anhydride were obtained. The esterification of 9-fluorenyl methanol with dithiodiacetic anhydride, 3,3′-dithiodipropionic anhydride and 4,4′-dithiodibutyric anhydride, respectively, was carried out overnight at room temperature under the condition of DMAP catalysis. Then, the intermediate products purified via column chromatography were activated with EDCI, HOBt and DMAP through an ice bath for 4 h. PPT was then added to the system and stirred for 48 h at room temperature to obtain three kinds of prodrugs. The purity of products could reach more than 97% after purification via pre-HPLC (LC3050N, Beijing Tong Heng Innovation Technology Co., Ltd., Beijing, China). The mobile phase was acetonitrile/water = 80:20. 

### 2.3. Preparation and Characterization of Prodrug NPs

The prodrug NPs were all prepared via the one-step nano-precipitation method. Briefly, FAP (1 mg) and DSPE-PEG_2k_ (0.25 mg) were dissolved in tetrahydrofuran (THF, 500 μL). Then, the mixed solution was added dropwise into deionized water (4 mL) under stirring (1000 rpm). Finally, the THF in the system was removed under vacuum conditions at 30 °C. The preparation of the other two PEGylated prodrug NPs (FBP NPs and FGP NPs) was similar. 

The size and zeta potential of prodrug NPs were determined with a Zetasizer (NanoZS, Malvern Co., Worcestershire, UK). Transmission electron microscopy (TEM, JEOL 100CX II, Kitakyushu, Japan) was used to observe the morphology of FAP NPs, FBP NPs and FGP NPs stained with 2% phosphotungstic acid. 

### 2.4. Molecular Docking Simulation

The assembly situations of three prodrug structures were simulated through molecular docking (Guangzhou Yinfu Information Technology Co., Ltd., Guangzhou, China), and the interactions between prodrug molecules were analyzed.

### 2.5. Colloid Stability

To monitor the colloid stability of prodrug NPs, FAP NPs, FBP NPs and FGP NPs were incubated in PBS and RPMI 1640 medium (with 10% Fetal bovine serum (FBS)), respectively, in a 37 °C shaker, and samples were taken at 1, 2, 4, 6, 8 and 12 h to measure the particle size. In addition, the sizes of FAP NPs, FBP NPs and FGP NPs were measured at 0, 3, 7, 15, 30, 45, 60 and 90 days, storing them in 4 °C atmospheres. The sizes of NPs were measured with a Zetasizer (NanoZS, Malvern Co., UK).

### 2.6. In Vitro Reduction-Responsive Release

The release behaviors of prodrugs in the presence of 0, 2, 5 and 10 mM DTT were investigated, respectively. As for the release medium, 30% ethanol was added to PBS (pH = 7.4) to make sure that hydrophobic PPT released from prodrug NPs could be fully dissolved in the system for easy detection. The released PPT was used for detection via HPLC (Acchorm S3000, Hitachi Instruments Co., Ltd., Shanghai, China) for 1, 2, 4, 6, 8 and 12 h after the addition of prodrugs. The proportion of binary eluents was methanol/water = 55:45. Meanwhile, the absorption wavelength was set at 295 nm.

### 2.7. Cell Culture

The mouse breast cancer cell line (4T1) and mouse embryo fibroblasts cell line (3T3) were provided by the American Type Culture Collection (ATCC). 4T1 cells were cultured in RPMI 1640 medium, and 3T3 cells were cultivated in DMEM-F12; both of them were mixed with 10% FBS, penicillin (100 units/mL) and streptomycin (100 μg/mL). Further, the cells were incubated at 37 °C in a proper atmosphere (5% CO_2_).

### 2.8. Cellular Uptake

Coumarin-6 (C-6)-labeled prodrug NPs were used to investigate the cellular uptake. The 4T1 tumor cells were planted in 24-well plates. After incubating with C-6 solution and C-6-labeled prodrug NPs with an equivalent concentration of C-6 (250 ng/mL) for 0.5 or 2 h, the medium containing drugs was abandoned. The cells were washed three times using cold PBS and then fixed with 4% paraformaldehyde. Moreover, the nucleus was stained with Hoechst 33342, and the samples were observed via confocal laser scanning microscopy (CLSM, C2, Nikon, Tokyo, Japan). In addition, 4T1 cells were seeded in 12-well plates (1 × 10^5^ cells/well). After cultivation for 12 h, fresh culture media containing FAP@C-6 NPs, FBP@C-6 NPs, FGP@C-6 NPs or C-6 solution (all in equivalent concentrations of coumarin, 250 ng/mL) were added to replace the precedent culture medium. After incubation for 0.5 or 2 h, the drug-containing culture medium was discarded and cells were washed with cold PBS three times. Then, the cells were collected, and the intracellular fluorescence signals were estimated via flow cytometry (BD FACSCalibur, BD Biosciences, San Jose, CA, USA).

### 2.9. Cytotoxicity

4T1 tumor cells were used to investigate the cytotoxicity of FAP NPs, FBP NPs, FGP NPs and PPT solutions by performing an MTT assay. 4T1 cells (2 × 10^3^ cells/well) were cultured in 96-well plates for 12 h. The precedent medium was replaced with fresh culture medium containing a series of concentrations of prodrug NPs and PPT solution and sequentially cultured for 48 h and 72 h. Then, 25 μL of MTT (5 mg/mL) was added to the culture media and incubated for another 4 h at 37 °C. Eventually, the media with MTT were gingerly discarded, and 200 μL of DMSO was added. After shaking for 10 min, absorbance was measured using a Varioskan LUX multimode microplate reader (Thermo Scientific, Waltham, MA, USA).

### 2.10. Animal Studies

Balb/C mice and Sprague–Dawley (SD) rats were used according to the requirements and regulations of the Animal Ethics Committee of Shenyang Pharmaceutical University. 

### 2.11. Pharmacokinetics Studies

SD rats (male, *n* = 6) were randomly assigned and given an i.v. dose of FAP NPs, FBP NPs, FGP NPs or PPT solution at 5 mg/kg (in equivalent concentrations of PPT) in sequence. At 0.03, 0.083, 1, 2, 4, 8, 12 and 24 h, blood was withdrawn from the eye socket, and the plasma was separated for analyses. Concentrations of prodrugs and PPT were quantified using LC-MS (Waters Xevo-TQD), respectively.

### 2.12. Biodistribution

DiR-labeled prodrug NPs were prepared to investigate the biodistribution of prodrug NPs in 4T1-tumor-bearing Balb/C mice. When the tumor volume reached 500 mm^3^, FAP@DiR NPs, FBP@DiR NPs, FGP@DiR NPs and DiR solution were injected through the tail vein (the DiR equivalent dose was 1 mg/kg). At 2, 4, 6, 8, 12 and 24 h after administration, the anesthetized tumor-bearing mice were photographed using an IVIS spectrum small-animal in vivo imaging system. In order to observe the distribution of drugs in major organs, 24 h after administration, the tumor-bearing mice were sacrificed, and the heart, liver, spleen, lung, kidney and tumor were dissected and photographed using an IVIS spectrum small-animal in vivo imaging system.

### 2.13. Maximum Tolerated Dose and Hemolysis Test

In order to investigate the biosafety of prodrug NPs, two doses of 10 mg/kg, 20 mg/kg and 30 mg/kg (equivalent dose of PPT) were set, and healthy mice were administered through tail intravenous injection every two days (4 times in total), The body weight change and survival curve were recorded during the treatment. 

In addition, the hemolysis ability of NPs and the solution was detected. We added prodrug NPs and PPT solution (equivalent dose of PPT) to the same amount of red blood cell suspensions, using normal saline and deionized water as a negative control and positive control, respectively. After 2 and 4 h of incubation in 37 °C shakers, the absorbance value at 545 nm was measured using a Varioskan LUX multimode microplate reader (Thermo Scientific, USA). The hemolysis rate (HR) was calculated according to the following formula: HR = (ODsample − ODnegative)/(ODpositive − ODnegative) × 100%.

### 2.14. In Vivo Antitumor Activity

4T1 tumor-bearing female Balb/C mice were modeled to investigate the in vivo antitumor activity of prodrug NPs. For this, 100 μL of 4T1 cell suspensions (5 × 10^7^ cells/mL) was injected subcutaneously into the right rear of BALB/C mice. When the tumor volume reached 100 mm^3^, the mice were randomly divided into 5 groups (*n* = 5): FAP NPs 20 mg/kg, FBP NPs 20 mg/kg, FGP NPs 20 mg/kg, PPT solution 10 mg/kg and control (saline). These formulations were administrated into the 4T1 tumor-bearing mice via tail intravenous injections at an interval of two days for four injections total, respectively. The tumor volume and body weight were measured and recorded once a day. Two days after the last dose, all mice were sacrificed, tumors were collected and soaked in 4% paraformaldehyde to preserve them, and they were prepared for staining with H&E. Tumor sections stained using the terminal deoxynucleoitidyl transferase dUTP nick end labeling (TUNEL) assay were obtained to evaluate damage caused by drugs to tumor cells.

### 2.15. Statistical Analysis

All data were calculated and presented as the mean ± SD. The significant differences between groups were identified by performing a *t*-test or one-way analysis of variance (ANOVA), and *p* < 0.05 was considered statistically significant.

## 3. Results and Discussions

### 3.1. Design and Synthesis of Prodrugs

As illustrated in [Fig pharmaceutics-15-00784-ch001], three prodrugs, consisting of diverse disulfide bonds as linkages (FAP, FBP, and FGP), were synthesized. Synthetic routes are shown in Appendix A. The chemical structures of three disulfide bond-bridged PPT prodrugs were confirmed via ^1^H nuclear magnetic resonance (^1^H NMR) and mass spectra (MS) analysis (Appendix A).

### 3.2. Preparation and Characterization of Prodrug NPs

Three prodrug NPs were prepared via the one-step nano-precipitation method. The appearance of the preparations showed obvious opalescence. The various parameters of the formulations are shown in Appendix A. Through measurements with the Malvern Zetasizer and observations of transmission electron microscopy, the particle sizes of the three prodrug NPs were about 100 nm, and the morphology was uniform (Figure 1A). Moreover, the zeta potential of the formulations was located at about −20 mV (Appendix A), which also indicates that there was a more suitable electrostatic gravity-repulsion force between particles, and the kinetic stability of the preparations was satisfactory. 

### 3.3. Molecular Docking Simulation

In order to investigate the self-assembly mechanism of prodrug NPs, molecular docking simulation of three prodrug compounds was performed. The simulation results of the three prodrugs were basically similar. Not only hydrophobic forces, but also π-π stacking, was proven to play an important role in the assembly process. This is due to the existence of conjugated systems that can provide π electrons in both Fmoc molecules and PPT molecules. In the process of assembly, forces interact with each other or themselves. 

### 3.4. Colloid Stability

The prodrug NPs were expected to remain stable in the blood circulation. For this purpose, the particle size changes of NPs in PBS and RPMI 1640 cell medium (containing 10% FBS) in 37 °C shakers were investigated to assess the stability of NPs by simulating an in vivo microenvironment. The particle size of the NPs did not change significantly within 12 h (Appendix A). In addition, the long-term stability of NPs in the storage process is also crucial. Therefore, particle size changes of the NPs during 90 days in the 4 °C refrigerators were regarded, and we found that after long-term storage at a low temperature, the particle size and the polydisperse index (PDI) of the NPs did not change significantly (Appendix A). In a word, NPs can remain relatively stable both in the simulated in vivo environment and the long-term storage process. 

### 3.5. In Vitro Reduction-Responsive Release

The reduction-responsive ability of a disulfide bond is a vital theoretical basis for our design of tumor-specific prodrugs. The introduction of a disulfide bond endowed the prodrugs tumor-specific response ability. In order to investigate the tumor-specific response release ability of three prodrug NPs, prodrug NPs were added into release medium containing DTT to mimic the high GSH in tumor cells, and the content of PPT in the system at different time points was detected via HPLC to calculate the percentage of prodrug release (Figure 2). To our surprise, the three prodrugs showed completely different release behaviors. The three prodrugs were almost not released without DTT. Once DTT was added into the system, FAP NPs showed an extremely sensitive reduction-response ability, releasing more than 80% of PPT within 2 h, while FGP NPs released slightly slower, and FBP NPs hardly released.

Based on our team’s previous work [24,45], the different release mechanism has been dissected (Figure 3). In the presence of DTT, the disulfide bond was cleaved, and the hydrophilic sulfhydryl group was exposed. Then, the cleavage of the ester bond was a key process for the release of PPT. The hydrophobic carbon chain length determines the trend in ester bond hydrolysis. The sulfhydryl group of FAP is only one carbon atom apart from the ester bond, so its ester bond is the easiest to be attacked and broken off. While FGP, for which the sulfhydryl group and ester bond are separated by three carbon atoms, should be the most difficult to hydrolyze among the three compounds, it happened that its carbon chain could form a relatively stable five-membered ring structure, which is a typical leaving group, thus the discrepancy of release rates was shown between FGP and FBP.

### 3.6. Cellular Uptake

4T1 cells were utilized to investigate the cellular uptake behavior of three prodrug NPs. C-6 was chosen to label the prodrug NPs for tracing. 4T1 cells incubated with C-6-labeled prodrug NPs or C-6 solution for 0.5 and 2 h were observed via CLMS (Figure 4A,B). The cellular uptake efficiency of prodrug NPs was significantly higher than that of C-6 solution, and the fluorescence intensity at 2 h was significantly higher than that at 0.5 h, which proved that the uptake was time-dependent. In addition, flow cytometry was used to detect cells after the same incubation as above, and it demonstrated that among the three groups of prodrug NPs, this was almost same, but was significantly higher than that in the C-6 solution group, and the time-dependence could be still observed (Figure 4C). The three prodrug NPs showed similar cell uptake efficiency, which could be ascribed to the similar size and surface properties of the three prodrug NPs. To sum up, prodrug NPs showed much better cellular uptake capacity than solution, which is the key to accurate and efficient drug delivery.

### 3.7. Cytotoxicity

The remarkable system-cytotoxicity of PPT has been reported in many studies, which is also a huge obstacle limiting its clinical application. We modified PPT with strong cytotoxic activity using a prodrug strategy and prepared self-assembled NPs. In order to investigate the cytotoxicity of prodrug NPs towards tumor cells, 4T1 cells were selected as the model for cytotoxicity tests, and the survival rate of 4T1 cells was measured by performing an MTT assay. IC_50_ values were calculated and are displayed in Appendix A. The three prodrug NPs showed attenuated cytotoxicity compared with PPT solution, probably due to the delayed release of PPT from prodrug NPs. In addition, FAP NPs showed higher cytotoxicity than FBP NPs and FGP NPs, which could be ascribed to faster rapid drug release (Figure 4D,E). As for the normal cells, 3T3 cells were selected to carry out the cytotoxicity assay under the same conditions. It was found that PPT solution exhibited strong cytotoxicity in 3T3 cells at a very low concentration as expected, while the cytotoxicity produced by prodrug NPs was all attenuated (Appendix A). In conclusion, the above results demonstrated that prodrug NPs have an ideal tumor-specific killing effect and alleviated the severe toxicity of PPT towards normal cells.

### 3.8. Pharmacokinetics Studies

SD rats were used to evaluate the pharmacokinetic behavior of prodrug NPs in vivo. Blood samples were collected at different time points within 24 h after i.v. administration, and plasma samples were analyzed to detect the contents of prodrug and PPT. Pharmacokinetic parameters were calculated and are presented in Appendix A. Plasma concentrations of the drug–time curve were plotted to show the pharmacokinetic behavior of the drugs in vivo. As shown in Figure 5, the AUCs of prodrug NPs were larger than that of PPT solution, indicating that prodrug NPs could effectively resist clearance by the reticuloendothelial system (RES) and the circulation time of drugs in vivo was prolonged, which will be beneficial for accumulation at the tumor site.

### 3.9. Biodistribution

4T1 tumor-bearing mice were constructed as the biodistribution model. After injecting DiR-labeled prodrug NPs or DiR solution, DiR fluorescence could be detected and observed through the in vivo imaging of small animals to trace the biodistribution of drugs in the body (Figure 6A). The fluorescence intensity at the tumor of prodrug NPs was higher than that in the DiR solution group at each time point, which could be ascribed to the better pharmacokinetic behavior of prodrug NPs. At 24 h after administration, mice were sacrificed, and the tumor and major organs were dissected out and photographed again (Figure 6B,C). The fluorescence intensity at the tumor site of the prodrug groups was significantly higher than that of the solution group.

### 3.10. Maximum Tolerated Dose and Hemolysis Test

Healthy Balb/c female mice were used to investigate the maximum tolerated dose (MTD) of prodrug NPs and PPT solution. The doses of 10, 20 and 30 mg/kg formulations were administered through the tail vein once every two days, four times (Figure 7A). The survival and weight changes of mice were recorded. When the dose was 10 mg/kg, the survival status of the formulation groups and the solution group were all good (Figure 7B), keeping the body weight stable (Figure 7D). However, when the dose of PPT reached 20 mg/kg, the toxicity of PPT solution became apparent, and death began to occur after the second dose (6 days) (Figure 7C). In addition, the body weights of the surviving individuals were significantly lower than those in the preparation groups (Figure 7E). Meanwhile, symptoms, such as diarrhea and hematochezia, persisted. When the dose reached 30 mg/kg, the mice died immediately after being injected with PPT solution. Survival improved in the preparation groups compared with that in the solution group, but there were still deaths during treatment (Appendix A). Therefore, the MTD was determined to be 20 mg/kg for preparation groups and 10 mg/kg for the solution group. Further, hemolysis tests were performed to reveal the destructive effects of prodrug NPs and PPT solution on erythrocytes (Figure 7F,G). When co-incubated with erythrocytes for 2 and 4 h, PPT solution was found to induce severe damage to erythrocytes, and hemolysis could be observed obviously, while the NPs resulted in no damage to them, which showed almost no difference from that in the negative controls. In a word, prodrug NPs indeed minimized adverse side effects of PPT.

### 3.11. In Vivo Antitumor Efficacy

4T1 tumor-bearing mice were constructed to assess the antitumor activity of the preparations in vivo, and the same administration regimen was adopted as above. When the tumor volume reached about 100 mm^3^, administration was started, and it was repeated once every two days for a total of four times (Figure 8A). Changes in tumor volume and body weight during treatment were recorded daily (Figure 8B,C). As can be seen from the figures, FAP NPs showed the strongest antitumor effect, and no significant weight loss was observed in all groups during the treatment. After the treatment, blood samples were collected from the mice, and they were subsequently sacrificed; then, the tumors were dissected out and weighed to calculate the tumor burden rate (Figure 8D). Compared with that in the saline group, the tumor burden rate of FAP NPs was remarkably reduced. Then, tumors were sectioned and stained for H&E and TUNEL (Figure 8E). In accordance with the results of the above experiments, FAP NPs stood out among all groups and showed extraordinary antitumor activity. Further, blood analysis was performed to investigate whether the liver and kidney functions were affected (Appendix A). As can be seen, all indicators of the treated mice were within the normal range. Finally, the heart, liver, spleen, lung and kidney sections of mice were also stained with H&E to evaluate whether the drug had toxic and side effects on major organs after treatment (Appendix A). There was no visible injury in major organs after administration.

## 4. Conclusions

In summary, three novel Fmoc-PPT prodrugs were designed by utilizing different lengths of disulfide bonds. As we hypothesized, the introduction of Fmoc improved the assembly ability, and the three PPT prodrugs could self-assemble into stable nano-systems. Thanks to the one-step nano precipitation method, the self-assembled prodrug NPs avoided the use of surfactants and cosurfactants. Then, prodrug NPs showed higher cell uptake efficiency than solution. In addition, the position of disulfide bonds in the linker also affected the reduction responsiveness of the prodrugs, thereby influencing the cytotoxicity. What makes sense is that our nano-delivery system mitigates the severe side effects of PPT solution by reducing the systemic toxicity and boosting the maximum tolerated dose. Finally, FAP NPs with an α-disulfide bond showed more-potent antitumor effects than FBP NPs and FGP NPs, which could be ascribed to higher reduction-responsiveness and faster tumor-specific drug release. We expect our study to make PPT a new step toward clinical application in cancer treatment.

## Data Availability

Not applicable.

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
