# Peer review of "Reduction-Hypersensitive Podophyllotoxin Prodrug Self-Assembled Nanoparticles for Cancer Treatment"

_pharmaceutics, 2023, doi:10.3390/pharmaceutics15030784_

Round 1

Reviewer 1 Report

The authors present the findings of a research study on the use of podophyllotoxin for cancer treatment. The authors address the challenges associated with using podophyllotoxin and describe their solution of designing and synthesizing novel prodrugs. These prodrugs showed promising results in reducing toxicity and increasing the tolerated dose, and the best antitumor effect was seen in prodrugs containing an α-disulfide bond. Overall, the paper is well-written and effectively demonstrates the results. Prior to publication, the following points should be addressed:

1.     Can you provide evidence for the reduction-responsive release mechanism of the prodrugs? It would be helpful to include a reference to previous work in the section on page 8 and include supporting data.

2.     Why do you think all three groups of prodrug NPs show the same uptake as detected by flow cytometry? An explanation should be added.

3.     Although the uptake of the three groups appears similar, the toxicology results show significant differences when comparing FGP NPs to the other two groups. Can you provide a release profile for the three groups to confirm if the release difference is the main reason for these toxicology differences?

Reviewer 2 Report

In this manuscript the authors present the design and preparation of three novel PTT-fluorene methanol prodrugs (PTT is podophyllotoxin) linked by different lengths of the disulfide bond, self-assemble into uniform nanoparticles (NPs), namely FAP NPs, FBP NPs, and FGP NPs. Then, they analyze the effect of the prodrugs in vitro and in vivo through cellular uptake, cytotoxicity, pharmacokinetics, biodistribution, maximum tolerated dose, and antitumor efficacy in vivo, showing the strong antitumor effect for the compound FAP NPs (with α-disulfide-bond) and lower for FGP NPs (g-disulfide-bond).

The presented results can contribute to the advancement of knowledge in more effective anticancer therapy for clinical application, and the work can be published after a minor revision. 

Below are the aspects that need to be reviewed:

- In the Introduction, Figure 1 is not specified;

- The writing of the label on axes and the TEM images in the inset in Figure 1 from the text must be enlarged to be able to clearly observe the NPs product;

- Also, in Figures 1A and B, the prodrugs should be noted;

- Labels and write from inset in Figure 2, Figure 3, Figure 4C,D,E, Figure 5, Figure 6, Figure 7B,C,D,E,G and Figure 8B,C,D need to be enlarged

Reviewer 3 Report

In this report, wang et al., designed and synthesized three novel PTT−fluorene methanol prodrugs linked by different lengths of disulfide bonds. Also, they showed all three PPT prodrugs could self-assemble into uniform nanoparticles (NPs) with high drug-loading capacity. They reported the lengths of disulfide bond affect the drug release, cytotoxicity, pharmacokinetic characteristics, in vivo biodistribution, and antitumor efficacy of prodrug NPs. However, the authors need to address the following concerns.

1.        The abstract is not written well and doesn’t reflect the complete results of the study. Can be rewritten?

2.        The drug release efficiencies of FAB, FBP, and FGP NPs at variable pH were not performed and discussed.

3.        Are there any changes observed in the blood profiles of mice observed post-administration of formulations? Did the formulation cause any signs of distress and aberrations in their growth patterns?

4.        The authors have not determined the IC50 values of the formulation. The inclusion of this data would provide more information about the dose administration.

5.        Did the authors perform any biocompatible studies with the formulations?

6.        What proportion do authors agree on this current approach would fit with existing standard therapies?

7.        The manuscript can be revised further for grammatical and typological errors.
